# A Cumulative Muscle Index and Its Parameters for Predicting Future Cognitive Decline: Longitudinal Outcomes of the ASPRA Cohort

**DOI:** 10.3390/ijerph18147350

**Published:** 2021-07-09

**Authors:** Ji-Yeon Baek, Eunju Lee, Woo Jung Kim, Il-Young Jang, Hee-Won Jung

**Affiliations:** 1Division of Geriatrics, Department of Internal Medicine, Asan Medical Center, University of Ulsan College of Medicine, Seoul 05505, Korea; complete_flow@naver.com (J.-Y.B.); eunjulee@amc.seoul.kr (E.L.); 2Department of Psychiatry, Yongin Severance Hospital, Yonsei University College of Medicine, Yongin 16995, Korea; woojay00@naver.com

**Keywords:** sarcopenia, cognitive impairment, sarcopenic parameters, gait speed

## Abstract

Sarcopenia and cognitive decline share the major risk factors of physical inactivity; previous studies have shown inconsistent associations. We aimed to identify the association of sarcopenia and its parameters with cognitive decline. The 3-year longitudinal outcomes of 1327 participants from the Aging Study of the Pyeongchang Rural Area (ASPRA) cohort were analyzed. Cognitive performance was evaluated using the Mini-Mental State Examination (MMSE), and sarcopenia was defined by the following: the original and revised Asian Working Group for Sarcopenia (AWGS), the original and revised European Working Group on Sarcopenia in Older People (EWGSOP), and the Cumulative Muscle Index (CMI), a novel index based on the number of impaired domains of sarcopenia. Approximately half of the participants showed meaningful cognitive decline. Sarcopenia by the original EWGSOP and the CMI were associated with cognitive decline. Only the CMI showed consistent predictability for cognitive impairment even with different criteria of the MMSE score (OR 1.23 [1.04–1.46]; OR 1.34 [1.12–1.59]; OR 1.22 [1.01–1.49], using the 1, 2, and 3 cut-off value, respectively). Of the CMI parameters, gait speed was satisfactorily predictive of 3-year cognitive impairment (OR 0.54 [0.30–0.97]). In conclusion, sarcopenia based on the CMI may be predictive of future cognitive impairment. Gait speed was the single most important indicator of cognitive decline.

## 1. Introduction

Dementia is a worldwide health issue, especially with the aging population. Currently, 47 million people are estimated to be living with dementia, and this number is anticipated to increase to 131 million by 2050 [1]. In South Korea, one of the fastest-growing countries, the prevalence of dementia in adults older than 65 years was approximately 10.2% in 2017; more than 2 million people are expected to have dementia in 2040 [2,3,4]. With the lack of curative pharmaceutical treatments, preventing dementia by controlling modifiable risk factors (less education, diabetes, hypertension, obesity, depression, smoking, and physical inactivity) has been considered crucial for alleviating the social impact of dementia [5,6,7]. According to a report, approximately 40% of dementia cases may be prevented by modifying these risk factors [6].

Among the risk factors, physical activity contributes to the largest proportion of dementia cases in the USA and Europe; it also impacts other risk factors such as diabetes, obesity, and depression [7]. Physical activity was inversely associated with the risk of Alzheimer’s disease [8] and was associated with improved global cognitive function in a meta-analysis of 39 randomized controlled trials [9]. Meanwhile, decreased physical activity is also considered a major pathophysiological risk for sarcopenia, a common geriatric condition characterized by decreased muscle mass, force, and/or physical performance [10]. Cross-sectional and longitudinal studies have shown the associations between sarcopenia and various adverse health outcomes such as functional decline, frailty, and mortality [11]. By sharing the major risk factors of physical activity, the association between the status of sarcopenia and future cognitive decline is predictable, even though previous studies have shown inconsistent associations [12,13,14].

Therefore, we aimed to investigate the association of sarcopenia and its parameters with cognitive decline among community-dwelling older people using a 3-year longitudinal cohort.

## 2. Materials and Methods

### 2.1. Study Population

In the study, we used the records of the Aging Study of the Pyeongchang Rural Area (ASPRA), a community-based prospective cohort study of Korean older adults. Established in 2014, the cohort study was designed to elucidate the natural course of frailty, geriatric syndromes, and disability in rural community-dwelling older people. All participants underwent annual comprehensive geriatric assessments, which included evaluations of medical conditions, physical activity, functional status, disability, cognitive function, mood, and nutritional status. The detailed descriptions of the initial study design and method have been demonstrated elsewhere [15].

From 2014, the regions covered by the study have been gradually expanded, and we included 1338 participants who were enrolled in the study between December 2014 and December 2016 for longitudinal analysis of 3-year cognitive changes. After excluding 11 individuals with missing baseline Mini-Mental State Examination (MMSE) findings, we analyzed the records of 1327 participants.

Among these, the 3-year longitudinal records were not available for 500 participants because of death (88), institutionalization (103), moving out (109), consent withdrawal (109), getting lost to contact (8), missing MMSE (79), and others (4). Therefore, the records of 827 participants were included in the final analysis (Figure 1). All participants provided written informed consent. The study was approved by the institutional review board of Asan Medical Center, Seoul, Korea (IRB No. 2015-0673) and was performed in accordance with the ethical standards laid out in the 1964 Declaration of Helsinki and its later amendments.

### 2.2. Assessment of Cognitive Performance

For cognitive performance, we used the Korean version of the MMSE, which uses scores ranging from 0 to 30 points. Cognitive dysfunction was indicated by MMSE scores of <24 [16]. In this study, meaningful cognitive decline was indicated by a decrease in the MMSE score of 1 or more points from the baseline to 3 years after the baseline [17]. As per this criterion, the study population was classified into two groups: no decline of MMSE and meaningful decline of MMSE. We also included ≥2 and ≥3 point reductions in the MMSE score in the sensitivity analysis.

### 2.3. Assessment of Sarcopenia

Muscle mass was assessed using the bioelectrical impedance analysis (InBody 620; InBody, Seoul, Korea) with measuring frequencies of 5, 50, and 500 kHz. The device measured the impedance of four limbs of participants who fasted overnight to minimize inter-individual fluid balance. Appendicular skeletal muscle (ASM) was quantified by summing the lean mass of both arms and legs. ASM was adjusted using the square of the height (ASM/ht²) to compare the muscle masses of the participants. Grip strength was measured by a handgrip dynamometer (T.K.K 5401 Grip-D; Takei, Tokyo, Japan) in the sitting position with the elbow flexed. The maximum of the two measurements in the dominant arm was recorded. For gait speed, the participants were asked to walk for a distance of 7 m at their usual walking speed on a flat indoor surface. After excluding the acceleration and deceleration intervals of 1.5 m each, the trained nurses measured the transit duration for 4 m using a digital stopwatch.

Sarcopenia was defined as low muscle mass with a low handgrip strength or slow gait speed by the original and revised Asian Working Group for Sarcopenia (AWGS) [18,19] and the original European Working Group on Sarcopenia in Older People (EWGSOP 1) [20]. In the revised EWGSOP (EWGSOP 2) guidelines, sarcopenia was defined as decreased muscle mass and decreased handgrip strength, and severe sarcopenia was defined as decreased muscle mass, decreased handgrip strength, and slow gait speed [11]. The Cumulative Muscle index (CMI), ranging from 0 to 3, was based on the number of impaired domains of sarcopenia among the following: decreased muscle mass, decreased handgrip strength, and slow gait speed [21].

Based on the sarcopenia definitions, different cutoffs for each parameter (low muscle mass, low handgrip strength, and slow gait speed) were used. In the original AWGS and CMI definitions, a low muscle mass was defined as an ASM/ht² of <7.0 kg/m² for men and <5.7 kg/m² for women. Low grip strength was defined as a maximal grip strength of <26 kg for men and <18 kg for women. Slow gait speed was defined as a speed of <0.8 m/s for both men and women [18]. In the revised sarcopenia definition by AWGS, a low muscle mass was defined as an ASM/ht² of <7.0 kg/m² for men and <5.7 kg/m² for women. Low grip strength was defined as a maximal grip strength of <28 kg for men and <18 kg for women. Slow gait speed was defined as a speed of <1.0 m/s for both men and women [19]. In both EWGSOP 1 and EWGSOP 2 definitions, a low muscle mass was defined as an ASM/ht² of <7.0 kg/m² for men and <5.5 kg/m² for women. Low grip strength was defined as a maximal grip strength of <27 kg for men and <16 kg for women. Slow gait speed was defined as a speed of <0.8 m/s for both men and women [11,20].

### 2.4. Assessment of Other Geriatric Parameters

Disability was assessed using validated scales for Koreans: the Korean versions of the activities of daily living (ADL) and the instrumental activities of daily living (IADL). The ADL scale is used to assess dependence based on 7 items (bathing, dressing, eating, toileting, continence, transferring bed or chair, and washing face and hands). The IADL scale is used to evaluate dependence based on 10 items (preparing meals, household chores, going out a short distance, grooming, handling finances, taking medications, shopping, laundry, using public transportation, and making phone calls) [22]. Multimorbidity was defined as having more than 2 of the 11 physician-diagnosed illnesses (diabetes, hypertension, angina, heart attack, congestive heart failure, stroke, kidney disease, asthma, chronic lung disease, arthritis, and cancer). Polypharmacy was defined as taking more than 5 prescribed medications. Nutritional status was evaluated using the Mini-Nutritional Assessment Short Form (MNA-SF), and a score of less than 11 points was associated with an increased risk of malnutrition [23]. Depressive moods were evaluated using the Korean version of the Center for Epidemiological Studies—Depression Scale (CES-D), ranging from 1 to 60 points. Depression was defined as a CES-D score of ≥21 [24].

### 2.5. Statistical Analysis

The categorical and continuous variables representing the baseline characteristics of the two groups were compared using the chi-squared test and *t*-test, respectively. Linear regression (continuous outcomes) and logistic regression (binary outcomes) were used to assess the longitudinal associations of the sarcopenia indices and parameters with the change in the MMSE score. Two different models were used for the analysis. In model 1, age, sex, and baseline MMSE score were adjusted, and in model 2, age, sex, baseline MMSE score, multimorbidity, ADL disability, and IADL disability were adjusted.

All analyses were conducted using STATA 15.0 (StataCorp, College Station, TX, USA). We considered two-sided *p* values < 0.05 as statistically significant.

## 3. Results

### 3.1. Baseline Characteristics

Of 827 participants who took part in the final analysis, 420 (50.8%) participants showed a 3-year meaningful decline of MMSE. The total MMSE score was higher in the meaningful MMSE decline group (mean 26.3, standard deviation [SD] 3.6) than in the no meaningful decline group (mean 25.0, SD 3.7) (Table 1). The distribution of the baseline MMSE scores and the 3-year changes in MMSE scores are demonstrated (Figure 2). The participants in the meaningful MMSE decline group were older (73.9 years, SD 5.5 years) than those in the no meaningful decline group (73.1 years, SD 5.4 years). The likelihood of IADL impairment was higher in the group of meaningful MMSE decline (35.2%) than in the group of no meaningful decline of MMSE (24.8%). There were no differences in the other sociodemographic factors (sex, education), prevalence of multimorbidity, muscle mass, muscle function (gait speed, grip strength), and the prevalence of other geriatric conditions (ADL impairment, polypharmacy, risk of malnutrition, depression) (Table 1).

### 3.2. Sarcopenia Status and Cognitive Decline

Of the different sarcopenia definitions (original AWGS, revised AWGS, EWGSOP 1, EWGSOP 2, and CMI), only those classified by the EWGSOP 1 and the CMI were associated with the magnitude of longitudinal decline in cognitive function measured by the MMSE (Table 2).

Further longitudinal analyses were performed using different criteria for a meaningful decline in the MMSE score from 1 to 3. In the analysis in which a reduction of 1 point in the MMSE score was considered a meaningful change, sarcopenia status by the EWGSOP 1 (odds ratio [OR] 1.46, confidence interval [CI] 1.05–2.04) and the CMI (OR 1.25, CI 1.06–1.48) predicted a decline in cognitive function after adjusting for age, sex, and baseline MMSE score. Predictability was maintained for sarcopenia status by the EWGSOP 1 (OR 1.43, CI 1.02–2.00) and the CMI (OR 1.23, CI 1.04–1.46) even after adjusting for age, sex, total MMSE score, multimorbidity, ADL disability, and IADL disability. However, only CMI predicted a decline in cognitive function when regarding a reduction of 2 or 3 points as meaningful change (OR 1.34, CI 1.12–1.59; OR 1.22, CI 1.01–1.49, respectively) (Figure 3).

### 3.3. Sarcopenia Parameters and Cognitive Decline

Of the specific parameters of sarcopenia (muscle mass index, handgrip, gait speed), only gait speed was correlated with a decline in cognitive function in model 1 and model 2, when the parameters were introduced as continuous variables. The decline in cognitive function was predicted using only baseline gait speed (OR 0.54, CI 0.30–0.97) after adjusting for age, sex, total MMSE score, multimorbidity, ADL disability, and IADL disability (model 2) (Table 3).

## 4. Discussion

In this study, half of the participants showed meaningful cognitive decline over 3 years. This cognitive impairment was best predicted by the CMI, among the currently available sarcopenia definitions. The CMI consistently showed a meaningful association with cognitive decline not only in the continuous outcome analysis but also in the binary outcome analysis. Moreover, predictability was maintained even when different MMSE criteria were used. Of the CMI parameters, the single most important factor was gait speed. To the best of our knowledge, it is the first sarcopenic index that could be used for predicting and evaluating future cognitive impairment in older adults.

There have been numerous inconsistent data on the association between sarcopenia and cognitive function [12,13,25]. Van Kan et al. reported that different operational definitions of sarcopenia indicated a null association with cognitive impairment [12], whereas Chang et al. considered sarcopenia as an independently relevant risk factor for cognitive impairment in the meta-analysis [13]. Kim et al. reported that sarcopenia was associated with cognitive decline only in men in a cross-sectional study [25]. Furthermore, when specific sarcopenic parameters including muscle mass and physical function (hand grip strength, gait speed) were analyzed separately, low muscle strength and gait speed rather than muscle mass could better predict cognitive impairment in two recent studies [26,27]. Meanwhile, only gait speed, and not hand grip strength or muscle mass, was reported to correlate with cognitive impairment [25].

In our study, the EWGSOP 1 definition of sarcopenia and the CMI were associated with reductions in the values of continuous variables of the MMSE (Table 2). However, only the CMI showed consistent predictability of cognitive decline for all three criteria for the MMSE scores (Figure 3). Sarcopenia, according to the EWGSOP 2 definition based on decreased muscle mass and hand grip strength but not gait speed, did not show any relevant association with cognitive decline. In a previous study, the CMI was highly associated with frailty and was more predictive of composite outcomes (death and hospitalization due to functional decline) than other pre-existing definitions of sarcopenia [21]. The difference between the CMI and the pre-existing sarcopenia definitions is the way of defining sarcopenia. Previous definitions evaluate sarcopenia dichotomously (no sarcopenia or sarcopenia), whereas the CMI uses serial numbers (score 0–3) to stratify the incremental sarcopenic burden [21]. Although slow gait speed was associated with cognitive impairment, hand grip strength also showed a trend of positive correlation with the MMSE score in our study. Therefore, the good predictability of the CMI may be attributed to this stratified grading system, which may allow equal considerations of all parameters of sarcopenia.

Gait is a complex motor task that demands bilateral coordination and postural control. Furthermore, all these movements are regulated sophisticatedly by the integration of multiple cognitive domains in the brain [28,29]. Of note, gait speed has received increasing attention as a predictive indicator of cognitive impairment and dementia [30,31,32]. As gait speed preceded cognitive decline, gait speed could detect a transition from mild to severe cognitive decline [30]. Furthermore, the concomitant decline in cognition and gait speed was accompanied by a 3-fold risk of developing dementia [31]. Similar to those of previous studies, our results showed that gait speed was the most meaningful parameter of sarcopenia (Table 3). Several studies have investigated the interplay between gait and cognitive function [28,29,33,34,35]. White matter hyperintensities in the periventricular region and subcortex and focal brain atrophy were associated with gait impairment and cognitive decline [28,34]. Crosslinking between the specific domain of cognition and gait measurements were also suggested. Executive function and attention were linked with almost all gait measures, including gait speed [29,35]. Also, memory and visuospatial ability were linked with specific gait measures such as gait speed, stride length, and step count [29,35].

While evaluated as predictors of future cognitive decline, CMI and gait speed also had respective advantages and disadvantages. Though gait speed showed best predictability and could be evaluated with ease and simplicity, it might not be related to sarcopenia status. For instance, the solitary problem of certain domains of the brain, which is associated with gait speed, might not cause definite cognitive decline and may not have any relationship with sarcopenia. Recently, studies have reported that pre-existing sarcopenia definition might not be applicable to the population with locomotor diseases such as spinal stenosis or herniated spinal disc, as low gait speed is necessarily concomitant with these diseases [36,37]. On the other hand, though calculating CMI could be more time-consuming than gait speed, it may overcome these drawbacks that a single gait speed has. Therefore, it is important to recognize that gait speed per se may not represent a sarcopenia status, and CMI could also fail to reflect sarcopenia status, since it counts gait speed as one of the sarcopenic parameters.

To the authors’ knowledge, our study first devised and successfully verified the CMI for predicting future cognitive decline using currently available sarcopenic parameters. Our well-designed cohort also had strength in terms of encompassing a wide range of sarcopenic phenotypes, as we recruited more than 95% of residents in the study region. Since the cohort region was isolated territorially and most of the residents were registered and cared for by the Korean government health care system, a high participation rate was feasible [15]. However, our study has limitations. Firstly, participants in the ASPRA cohort had fewer years of education, although the sociodemographic characteristics of the ASPRA cohort were representative of those of the Korean rural population [15]. Second, inflammation status or insulin resistance was not studied due to the lack of blood samples. Third, the clinical findings of the MMSE decline, such as progression to dementia, were not considered in this study. Fourth, only MMSE was used for assessment of cognitive function. Since MMSE has a relatively lower sensitivity in detecting mild cognitive impairment compared with other cognitive assessment tools such as the Montreal Cognitive Assessment (MoCA) [38], participant’s elaborate changes of cognitive function might not be detected in MMSE scores.

## 5. Conclusions

A newly derived sarcopenic index, the CMI, may be predictive of future cognitive impairment. Of the CMI parameters, gait speed was the single most important indicator of cognitive decline in community-dwelling older adults. Future research should consider further validation of CMI in other cohorts, and the clinical diagnoses of cognitive decline, such as mild cognitive impairment or dementia. It will also be interesting to modify the CMI grading system to use the weighted value of each sarcopenia parameter to evaluate the degree of correlation with cognitive impairment.

## Figures and Tables

**Figure 1 ijerph-18-07350-f001:**
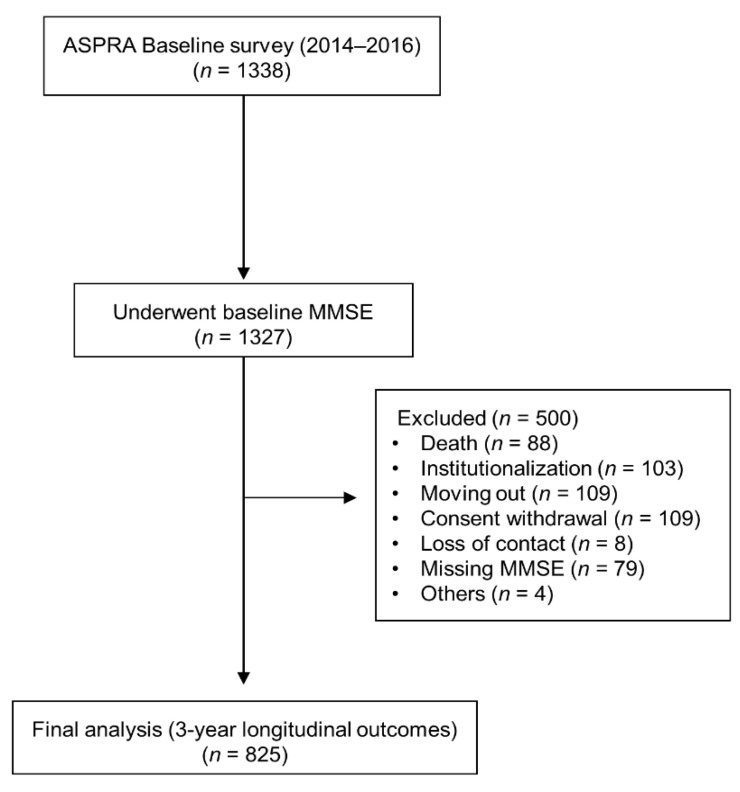
Flow chart of the study population.

**Figure 2 ijerph-18-07350-f002:**
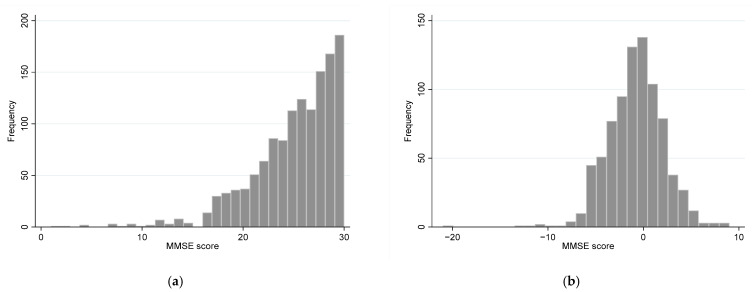
Histogram of MMSE score. (**a**) Distribution of the baseline MMSE score (*n* = 1327); (**b**) Three-year changes in MMSE score (*n* = 825). MMSE, Mini-Mental Status Examination.

**Figure 3 ijerph-18-07350-f003:**
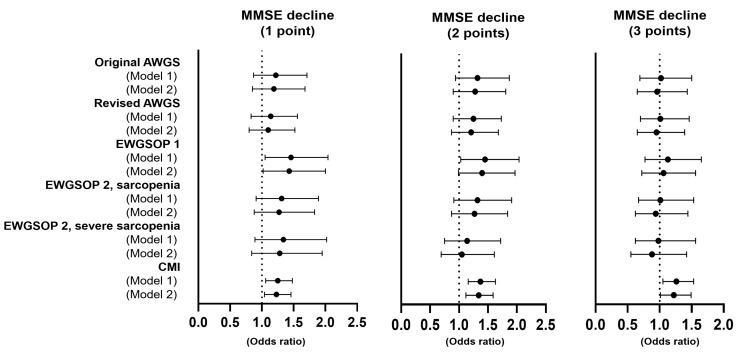
Longitudinal associations of sarcopenia with meaningful decline in the MMSE score. Model 1 adjusted for age, sex, and the baseline MMSE score; Model 2 was adjusted for age, sex, baseline MMSE score, multimorbidity, ADL disability, and IADL disability. MMSE, Mini-Mental Status Examination; AWGS, Asia Working Group Society; EWGSOP, European Working Group on Sarcopenia in Older People; CMI, Cumulative Muscle Index; CI, confidence interval.

**Table 1 ijerph-18-07350-t001:** Demographic characteristics of the population.

	No Decline of MMSE	Meaningful Decline of MMSE	*p*
Number	407 (49.2%)	420 (50.8%)	
Age	73.1(5.4)	73.9(5.5)	0.025
Sex (male)	169 (41.5%)	169 (40.2%)	0.707
Education level (years)	5.0 (3.2)	5.0 (3.2)	0.948
Multimorbidity	158 (38.8%)	165 (39.3%)	0.891
Hypertension	218 (53.6%)	232 (55.2%)	0.629
Diabetes	69 (17.0%)	71 (16.9%)	0.985
Stroke	12 (3.0%)	9 (2.0%)	0.462
Body mass index (kg/m²)	24.8 (3.5)	24.7 (3.4)	0.776
Muscle mass index (kg/m²)	6.5 (1.2)	6.4 (1.2)	0.486
Gait speed (m/s)	0.81 (0.30)	0.78 (0.28)	0.112
Grip strength (kg)	23.2 (9.5)	22.1 (9.7)	0.099
ADL impairment	43 (10.6%)	43 (10.2%)	0.878
IADL impairment	101 (24.8%)	148 (35.2%)	0.001
Polypharmacy	79 (19.4%)	104 (24.8%)	0.064
Risk of malnutrition	78 (19.2%)	86 (20.5%)	0.636
Depressive mood	28 (6.9%)	34 (8.1%)	0.507
MMSE score	25.0 (3.7)	26.3 (3.6)	<0.001

Values are presented as the mean ± standard deviation or number (%). MMSE, Mini-Mental Status Examination; ADL, activities of daily living; IADL, instrumental activities of daily living.

**Table 2 ijerph-18-07350-t002:** Longitudinal associations of sarcopenia with cognitive function measured by the MMSE.

	Linear Regression Model 1		Linear Regression Model 2	
	Coefficients (SE)	*p*	Coefficients (SE)	*p*
Original AWGS	−0.36 (0.22)	0.107	−0.31 (0.22)	0.163
Revised AWGS	−0.27 (0.21)	0.207	−0.21 (0.21)	0.316
EWGSOP 1	−0.64 (0.22)	0.004	−0.58 (0.22)	0.009
EWGSOP 2				
Sarcopenia	−0.32 (0.24)	0.181	−0.25 (0.24)	0.299
Severe sarcopenia	−0.42 (0.27)	0.127	−0.30 (0.27)	0.265
CMI	−0.41 (0.11)	<0.001	−0.36 (0.11)	0.001

Model 1 was adjusted for age, sex and the baseline MMSE score; Model 2 was adjusted for age, sex, baseline MMSE score, multimorbidity, ADL disability, and IADL disability. MMSE, Mini-Mental Status Examination; AWGS, Asia Working Group Society; EWGSOP, European Working Group on Sarcopenia in Older People; CMI, Cumulative Muscle Index; SE, standard error.

**Table 3 ijerph-18-07350-t003:** Longitudinal associations of sarcopenia parameters with cognitive decline measured by the MMSE.

	Linear Regression Model 1		Linear Regression Model 2		Multivariate Logistic Regression Model 1	Multivariate Logistic Regression Model 2
	Coefficients (SE)	*p*	Coefficients (SE)	*p*	Odds Ratio (95% CI)	Odds Ratio (95% CI)
Muscle mass index	0.13 (0.10)	0.207	0.09 (0.11)	0.418	0.98 (0.84–1.15)	1.01 (0.86–1.19)
Hand grip	0.03 (0.02)	0.108	0.03 (0.02)	0.078	0.98 (0.95–1.00)	0.97 (0.95–1.00)
Gait speed	0.98 (0.38)	0.009	0.81 (0.38)	0.033	0.51 (0.28–0.90)	0.54 (0.30–0.97)

Model 1 was adjusted for age, sex, and the baseline MMSE score; Model 2 was adjusted for age, sex, baseline MMSE score, multimorbidity, ADL disability, and IADL disability. MMSE, Mini-Mental Status Examination; SE, standard error; CI, confidence interval.

## Data Availability

The data presented in this study are available on request from the corresponding author.

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
