# Peer review of "A Cumulative Muscle Index and Its Parameters for Predicting Future Cognitive Decline: Longitudinal Outcomes of the ASPRA Cohort"

_ijerph, 2021, doi:10.3390/ijerph18147350_

Round 1

Reviewer 1 Report

This manuscript reports on the results of a well-conducted large prospective study conducted in Korea to appreciate factors possibly predicting cognitive outcome. A series of well-defined scores was used to evaluate sarcopenia and compare their baseline values with evolution of the mini mental state examination (MMSE). EWGSOP 1 and CMI appeared to be the most predictive of cognitive impairment. Among parameters evaluating sarcopenia, gait speed stood out as the most predictive of cognitive impairment.

The introduction, material and methods are well presented. Reading the results appears a little more confusing. Some of the tabulated data could be additionally summarized in a graphical fashion such as a forest plot.

The discussion could better discuss the respective advantages and disadvantages of the CMI vs. gait speed. Indeed, if the latter parameter really stands out as equally predictive as CMI, this means that it could easily be applied to large populations. Would the authors be able to determine a gait speed threshold associated with the risk of cognitive decrease (i.e. ROC curve)?

Minor

Supplementary figure deserves a better legend and axes annotation.

The reference list must be checked for consistency (only first word with first letter capitalized).

Reviewer 2 Report

This is a well-designed study that measures the correlation of sarcopenia and the risk of cognitive decline in elderly people. The strengths of the study are that it includes a large cohort of patients, uses well validated measures and have a good statistical analysis. Also, it is well written, and it is easy to read through the manuscript.

However, I have several concerns that should be addressed.

Major changes:

  • The authors do not state how is sarcopenia defined by the Cumulative Muscle Index. The SPS index (reference 21) measures the burden of sarcopenia, but it has no diagnostic value. Have the authors used a cut-off value of the index to define sarcopenia? This is not the purpose of the index.
  • It is true that gait speed has been proved in the study to be the best predictive factor of cognitive decline. However, it should be noted that an altered gait speed in isolation is not exclusively related to sarcopenia. Also, a low gait speed performance can be associated to an impaired motor control due to different cognitive domains that can be affected in several neurological diseases, as the authors state in the discussion. Thus, low gait speed alone is not a specific marker of sarcopenia. This should be well explained in the discussion.
  • In that sense, authors should also explain if the predictive value of the CMI index is exclusively related to the abnormal gait speed. Then, the index has limitations, as explained above.

Minor changes:

  • There is an imaging mistake in figure 1 that should be corrected.
  • In Line 100 it is stated: Sarcopenia was defined as low muscle mass with low handgrip strength or slow gait speed by the original and revised Asian Working Group for Sarcopenia (AWGS) [18, 19]. However, according to the references, sarcopenia is defined as low handgrip strength and slow gait speed. In case one of them is missing, a decreased muscle mass measurement is required.

Reviewer 3 Report

The work investigates the capacity of the CMI and the EWGSOP to predict cognitive impairment as measured by the MMNI. The reviewer acknowledges the large sample size that strengthen the conclusions and findings. The straight forward approach and both, continuous and binary regressions are also recognized. I include below a list of questions/suggestions that I recommend to address prior publication:

Minor comments

Method: mainly restructuration + clarifications

  • Study population – can you please indicate the cut-off criteria of the MMSE? Is it 24? Also, only 11 individuals out of 1338 were below it?
  • Missing one picture in Figure 1..
  • The reduction in MMSE of one point over time refers to the 3 years follow up? I see that this is mentioned later on in the text, I would suggest to explain it directly in the Method.
  • It is not clear the criteria that you use to create the subgroups; i.e. what is considered a meaningful MMSE decline? If -1poins is considered meaningful, I would suggest to rename this second subgroup as no “MMSE decline”.
  • In the same line, did you take into account the baseline when determining the meaningful MMSE decline? – it is not the same going from 30 to 29 than from 25 to 24? Are both equally considered as meaningful decline?

Results: suggestions + new model

  • I suggest to include the figure in supplemental materials directly in the text. It shows interesting information and it is only 1 more image.
  • To explore the relationship between the parameters, you created two models (models 1 and model 2) differently adjusted. This could go directly in the sections statistics to facilitate the reading.
  • Did you also adjust the MMSE decline by other parameters such as diabetes or depressive mood? – this is a manipulation interesting as there appear to be differences between the groups. This could be an interesting model 3.

Discussion

  • Can you explain a bit more the reasons why the MMSE was higher in the subgroup with the meaningful MMSE decline? That’s an interesting finding by itself.

Round 2

Reviewer 2 Report

The authors have made the corrections and recomendations suggested. The attached paragraphs need English language editing.

I believe this version of the manuscript is suitable for publication.